# A New Method for Hail Detection from the GPM Constellation: A Prospect for a Global Hailstorm Climatology

**Sante Laviola** [1,*], **Giulio Monte** [1], **Vincenzo Levizzani** [1], **Ralph R. Ferraro** [2] **and James Beauchamp** [3]

1   CNR-ISAC, via Gobetti 101, 40129 Bologna, Italy; g.monte@isac.cnr.it (G.M.); v.levizzani@isac.cnr.it (V.L.)
2   NOAA-NESDIS, University Research Court, College Park, MD 20740, USA; ralph.r.ferraro@noaa.gov
3   Earth System Science Interdisciplinary Center (ESSIC), Univsrsity of Maryland, College Park, MD 20742, USA; vajim@umd.edu
*   Correspondence: s.laviola@isac.cnr.it; Tel.: +39-051-639-8019

**Abstract:** A new method for detecting hailstorms by using all the MHS-like (MHS, Microwave Humidity Sounder) satellite radiometers currently in orbit is presented. A probability-based model originally designed for AMSU-B/MHS-based (AMSU-B, Advanced Microwave Sounding Unit-B) radiometers has been fitted to the observations of all microwave radiometers onboard the satellites of the Global Precipitation Measurements (GPM) constellation. All MHS-like frequency channels in the 150–170 GHz frequency range were adjusted on the MHS channel 2 (157 GHz) in order to account for the instrumental differences and tune the original model on the MHS-like technical characteristics. The novelty of this approach offers the potential of retrieving a uniform and homogeneous hail dataset on the global scale. The application of the hail detection model to the entire GPM constellation demonstrates the high potential of this generalized model to map the evolution of hail-bearing systems at very high temporal rate. The results on the global scale also demonstrate the high performances of the hail model in detecting the differences of hailstorm structure across the two hemispheres by means of a thorough reconstruction of the seasonality of the events particularly in South America where the largest hailstones are typically observed.

**Keywords:** hail detection; GPM constellation; hail climatology; passive microwave

## 1. Introduction

Hail detection is an open issue from the remote sensing point of view both from the ground and from space. Hail is extremely difficult to observe using passive and active sensing due to signal attenuation and the relatively scarce knowledge of the cloud structure in hailstorms. Several approaches have been recently proposed mainly using radar data from the ground in connection with observations in the visible and infrared from geostationary satellites (e.g., [1,2]).

However, the potential offered by the Global Precipitation Measurement (GPM) constellation (GPM-C) for monitoring precipitation and severe storms is unprecedented [3,4]. Since the late 1980s [5,6], the increasing number of passive microwave (PMW) satellite radiometers [7] has stimulated the development of a variety of algorithms for studying clouds and infer precipitation rates [8,9]. The high correlation between precipitation intensity and brightness temperature (TB) depression has been largely used as a diagnostic basis for identifying rain clouds and deriving hydrometeor phase. Spencer et al. [6] first correlated the signature from several severe storms over the US to the TB depression at 37 GHz mainly due to the scattering from large falling hail. These pioneering studies opened the way to the

retrieval of cloud properties using the signal of high-frequency microwave (MW) channels as a proxy to identify convective clouds and detect hail cores [10].

The response of the 19, 37, and 85 GHz channels on board the Tropical Rainfall Measurement Mission (TRMM) was used in predicting the probability of hail at the surface [11–13]. Cecil and Blankenship [14] used the frequencies at 36.5 and 89.0 GHz of the Advanced Microwave Scanning Radiometer for the Earth Observing System (AMSR-E) and the frequencies at 37.0 and 85.5 GHz of the TRMM Microwave Imager (TMI) to develop a climatology of severe hailstorms.

These studies paved the way to apply high frequency MWs to infer hail patterns. High-frequency channels offer new possibilities in observing precipitating clouds and retrieving key parameters. Leppert and Cecil [15] used airborne PMW data collected during intense convection episodes over Oklahoma to delineate the signal variation of the GPM Microwave Imager (GMI) frequency channels in recognizing the signature of various hydrometeors. Mroz et al. [16] expanded this approach by identifying the signatures by hail combining the measurements of the GMI with reflectivities from the Dual-frequency Precipitation Radar (DPR) on board the GPM Core Observatory (GPM-CO). They matched two-year simultaneous observations of the GPM with the US Next Generation Radar (NEXRAD) network data to catalog signals from hailstorms as a proxy for detecting hail. Ni et al. [17] defined two best-score thresholds at 37.0 GHz and Ku band and built a global distribution of possible hail events using two years of observations from the GPM-CO.

Using 16 years of TRMM TB-derived precipitation data paired with surface hail reports over the US, Bang and Cecil [18] calculated the probability curves in the 10–85 GHz frequency domain to fit GMI channel observations for the detection of hail patterns. The nearly global climatology of hail developed in this study has shown that the majority of hailstorms develops over South America and the central United States. Similar results were found by Ferraro et al. [19], who applied a threshold algorithm based on Advanced Microwave Sounding Unit (AMSU)-based data to derive hail occurrences and generate a global climatology of hailstorms over land. This investigation demonstrates the sensitivity of frequencies higher than 85 GHz while sampling a wide range of hail diameters.

The potential of the 90–190 GHz frequency range when employed in the classification of cloud types and in the detection of signals from different hail sizes was recently proved by Laviola et al. [20] and Ferraro et al. [21] extending previous approaches (e.g., [14]) to these frequencies that are now available on several platforms. MW high frequencies offer the advantage of very high sensitivity to the scattering signature from different ice particles with diameters from a few millimeters to 10s of centimeters. Thus, we are able to classify the region of convective clouds where different hail sizes are generated by identifying severity areas characterized by small ice aggregates potentially forming hail, large hail and super hail [20].

The AMSU-B/MHS (hereafter AMSU-B for Advanced Microwave Sounding Unit-B and MHS for Microwave Humidity Sounder) 150–157 GHz frequency is used to feed a novel probability-based model, the Microwave Cloud Classification method for Hail detection (MWCC-H), that associates probability values to signals from very small ice particles (low probability) to very large hailstones (d > 10 cm) typically marked by likelihoods close to 1.

The present study aims at extending the hail detection of the MWCC-H model proposed by Laviola et al. [20] to the entire GPM-C where several radiometers operate in the 150–170 GHz frequency range. All GPM-C radiometers with frequency channels in this range were empirically adjusted to the MHS channel at 157 GHz and then the MWCC-H model fitted to each instrument. In this way, the GPM-C can be considered as a single MW radiometer with the unprecedented capability of monitoring the evolution of hailstorm systems at very high temporal resolution. Hence, the rationale of this study is the creation of a satellite-borne dataset that makes available uniform information for investigating hail events from the local to the global scale using the GPM-C. An example is presented where the MWCC-H model generalized to the GPM-C allowed the reconstruction of the complete hailstorm sequence that affected the eastern Italian coast on the Adriatic Sea. Hence, a first global climatology attempt is presented.

Section 2 discusses the methodology used for fitting the original method for hail detection to the GPM-C radiometers and the calibrating datasets. Section 3 describes the detection of hail clouds from the GPM-C adopting the new hail detection algorithm. Section 4 shows a global scale application for studying the seasonality of hail patterns. Section 5 concludes and discusses future investigation areas.

## 2. Data and Methodology

The model for hail detection presented in Laviola et al. [20] and originally developed for MHS has been directly employed for sensing hail patterns from similar radiometers orbiting within the GPM-C (Figure 1). This first experimental test applied to the Advanced Technology Microwave Sounder (ATMS), the Special Sensor Microwave Imager/Sounder (SSMIS), and the GMI was useful to evaluate the sensitivity and flexibility of the method in switching the instrumental characteristics technically similar to the MHS (reference instrument) but was substantially different at the same time.

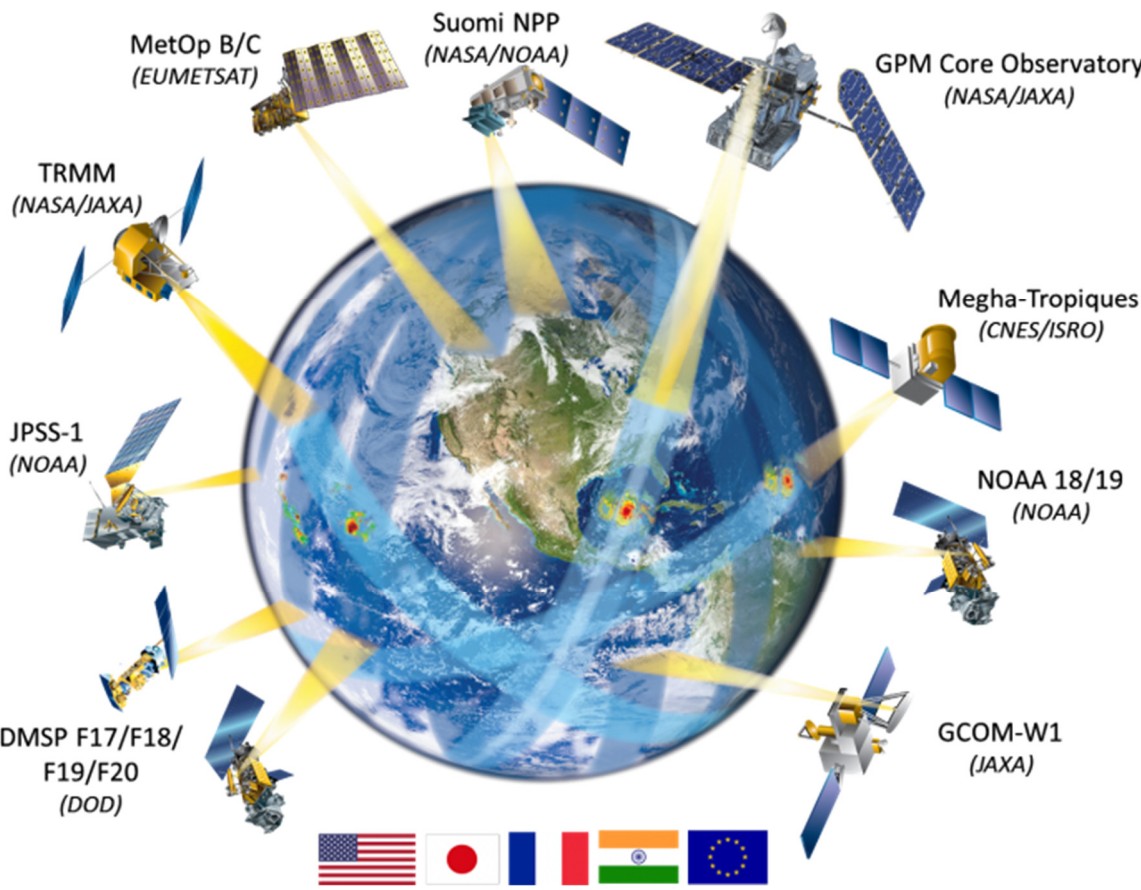

**Figure 1.** The Global Precipitation Measurements (GPM) constellation includes US and international satellite missions. The consortium comprises the European Organization for the Exploitation of Meteorological Satellites (EUMETSAT); the Japan Aerospace Exploration Agency (JAXA); the French Centre National d'Etudes Spatiales (CNES); the Indian Space Research Organisation (ISRO); the US Department of Defense, Defense Meteorological Satellite Program (DMSP); and the National Oceanic and Atmospheric Administration (NOAA). (Courtesy of NASA).

Although the first results showed an overall high performance of the method in identifying hail clouds, further applications showed the limitation of this crude implementation in quantifying the hail probability often overestimated when compared with simultaneous MHS overpasses. This behavior of the MWCC-H model was expected considering the technical differences between the GPM-C radiometers equipped with frequency channels in the 150–170 GHz range (hereafter MHS-like, see Table 1). Therefore, a better evaluation of the impacts of such differences has been undertaken.

To do that, we consider three key variables as a source of main displacement between the MHS-like and the MHS radiometers: (1) the sampling frequency, (2) the scan mechanism, and (3) the spatial resolution in terms of the instantaneous field of view (IFOV).

**Table 1.** List of the MHS-like radiometers used in this work compared with MHS characteristics (reference instrument).

| Satellite Mission | Instrument | Frequency (GHz) | Polarization | Scanning | Resolution (km) |
|---|---|---|---|---|---|
| GPM | GMI | 166.5 | V | Conical (CO) | $4 \times 7$ ($\approx$6 km) |
| DMSP-F17/F18/F19 | SSMIS | 150 | H | Conical (CO) | $13 \times 16$ ($\approx$14 km) |
| NPP/N20 | ATMS | 165.5 | QH | Cross-track (CT) | 16 km (nadir) |
| NOAA/MO | MHS | 157 | V | Cross-track (CT) | 16 km (nadir) |

Since the MHS-like radiometers have technical similarities with the reference MHS instrument (see Table 1), we conceived an ad hoc pairing "MHS-Others" with the aim of exploiting these similarities to evaluate the weight of the three key variables. This comparison strategy reveals a robust tool for adjusting each MHS-like sensor to feed the original hail detection model.

In Table 2, the differences of three key variables with respect to the MHS characteristics have been quantified. The MHS–ATMS pair, being both sensors cross-track scanning with the same spatial resolution at the ground, is a good tracer for evaluating the effect of the different sampling frequency. The MHS-SSMIS pair, which has the same frequency difference found between MHS and AMSU-B and a similar spatial resolution of the MHS at nadir, can be exploited to investigate the impact of the different scan mechanism. Finally, the MHS–GMI pair is used to weigh the impact of different spatial resolutions once the dependency from the other two variables has already been considered.

**Table 2.** Key variable differences between MHS (nadir view) and MHS-like radiometers: frequency ($v$), scanning geometry (scan), and instantaneous field of view (IFOV). The cross-track and conical instruments are identified with CT and CO, respectively.

| Satellite Mission | $\Delta v$ (GHz) | $\Delta$Scan | $\Delta$IFOV (km) |
|---|---|---|---|
| MHS-GMI | 9.5 | CT-CO | 10 |
| MHS-SSMIS | 7 | CT-CO | 2.0 |
| MHS-ATMS | 8.5 | CT-CT | 0 |

To evaluate the effects of these three key variables, a few case studies were selected. In order to improve the matching between each satellite pair a time-dependent criterion was used to force the selection of case studies. The hailstorm events used in this study were selected within a time lag of 30 min which is the maximum time distance between each satellite pair. This rather strict time constraint on one hand drastically reduces the number of hail events to be analyzed, but at the same time significantly mitigates the potential errors from the missing sampling of the same event due to the rapid evolution of the convective systems. On the basis of this time condition, one hailstorm for each satellite pair was identified. Table 3 shows the details of the selected events.

**Table 3.** Hailstorm events selected for defining the three satellite pairs. The hailstorm diameters are provided by the NOAA Storm Prediction Center (SPC) and the European Severe Weather Database (ESWD, [22]).

| Satellite Pairs | Location | Date | Hail Diameter (cm) | No. of Data |
|---|---|---|---|---|
| MHS-GMI | Italy (Naples) | 05 Set 2015 | 8.0 ~ 12.0 | 7739 |
| MHS-SSMIS | US (North Dakota) | 10 Jun 2017 | 2.5 ~ 5.0 | 4781 |
| MHS-ATMS | Italy (Sicily) | 22 Jan 2015 | 1.0 ~ 4.0 | 2634 |

The sampling frequency impacts the sensor sensitivity to scattering processes from ice particles in extinguishing the upwelling radiation. Generally, higher frequencies are more sensitive to the contribution of scattering to the total extinction of the radiation field reaching the satellite. As demonstrated by theoretical studies [23,24], cloud ice increasingly impacts high-frequency MWs as a function of ice amount and cloud top height. The sensitivity to scattering by ice particles significantly changes for frequency higher than 85 GHz by progressively depressing the TBs proportionally to the increasing frequency [10,15]. However, we verified that for hail clouds and in the 150–170 GHz frequency range, the TB reduction is within 5 K. By comparing quasi-simultaneous observations from MHS and ATMS in presence of hailstorms, we found that the frequency displacement between the two instruments ($\Delta\nu = 8.5$ GHz, as shown in Table 2) induced a maximum TB difference ($\Delta$TD) of 5 K. The use of the ATMS as a benchmark for quantifying the impact of different sampling frequencies is justified by the same characteristics of two instruments in terms of spatial resolution and scanning geometry such as the pair AMSU-B-MHS ($\Delta\nu = 7.0$ GHz; AMSU-B is the MHS predecessor). Therefore, by assuming a $\Delta$TD of 5 K, we assume no strong impact on the general performance of the MWCC-H model. This conclusion is supported by the results of Laviola et al. [20], whose diagram (their Figure 8) of the theoretical performance of the MWCC-H predicts a probability value from 0.02 to 0.05 when an increment from 1 K to 5 K is induced.

The scanning mechanism mainly affects the variability of the IFOV size on the scanline during each observation. Conically scanning sensors are equipped with a circular rotating antenna scanning a uniform cone at the surface slanted by unchanging incidence angles for a fixed suite of frequencies (45.0° for SSMIS at 150 GHz and 45.3° for GMI at 166.5 GHz). On the contrary, radiometers with a cross-track scanning mechanism are characterized by different IFOVs from the local spacecraft zenith to the edges of the scanning. As argued, satellite instruments with a different scanning mechanism differ more substantially in terms of surface spatial resolution than in the scanning mode. Therefore, even though the scanning mechanism will be evaluated as an independent variable, the high dependence on the spatial resolution at the ground has to be accounted for.

To evaluate the effect of the different scanning mechanisms, we exploit the technical similarities between MHS and SSMIS. As indicated in Table 2, the MHS-SSMIS pair shows the same frequency difference of the MHS-AMSU-B pair ($\Delta\nu = 7.0$ GHz) and a similar spatial resolution when MHS looks at the nadir ($\Delta$IFOV = 2 km). Hence, on the basis of selected simultaneous MHS-SSMIS observations, we try to assess the magnitude of the TB reduction and reconstruct its impact on the hail detection model. Our results show that SSMIS measures colder cloud regions than MHS showing $\Delta$TDs in the 15–20 K range corresponding to increments of hail probability values in the 0.08–0.12 range. When explaining these large discrepancies between MHS and SSMIS measurements, it should be noted that the matching was done with MHS scan positions far from the nadir. Thus, the variability of the IFOV largely controls the measurement of the radiation field by enhancing the TB reduction where the spatial resolution is higher. This consideration is corroborated by further analysis where the MHS–GMI pair is used to estimate the effect of spatial resolution in observing the same hail system. On the basis of the above discussion, in this matching we considered negligible the other differences in terms of sampling frequency and scanning mechanism.

Despite the low number of crossing overpasses, we have identified three hail events were MHS and GMI can be matched to evaluate the impact of spatial resolution in terms of scan position. As demonstrated by Bennartz [25], the size of the effective field of view (EFOV) of AMSU-B (MHS), which is the actual field of view (FOV) of the spaceborne sensor including effects of antenna rotation, changes in the cross-track direction from 20 to 65 km by moving from nadir (scan position 45) to the edges of the scanline (scan positions 1 and 90). Thus, GMI resolves each observation from 3 to 10 times better than MHS. Because hailstorms are very localized and evolve quickly in time, the spatial resolutions associated with the time distance between GMI and MHS observations are key factors. An additional variable is the microphysics of convection regulating the dynamics of the hail system and controlling the scattering process which extinguishes the signal to the satellite. Hence, even though

these three variables are strictly interconnected, we argue that microphysics and time distance are more strictly dependent: hailstorm microphysics rapidly changes the inner structure of the system, and then the response to the satellite changes correspondingly. A large time distance allows for the triggering of several evolution stages in the hailstorm so that two observations close to each other can seldom be considered simultaneous. This mostly justifies the observed displacement between GMI and MHS measurements.

Table 4 synthetizes the ΔTBs (negative values mean that the TBs from GMI are lower than those from the MHS) between GMI and MHS as a function of time distance Δt and scan position (i.e., spatial resolution in terms of divergence from the GMI) for the three selected hailstorms. ΔTBs progressively decrease more as a function of time distance than of the scan number which is symmetrically placed far from the MHS nadir position (MHS scans 90 FOVs).

**Table 4.** ΔTBs as a function of time lag (Δt) between GMI and MHS and MHS scan position.

| Hailstorm | Δt (min) | ΔTB (K) | MHS Scan |
|---|---|---|---|
| Italy (Naples) | 13 | −22.72 | 78 |
| Italy (Abruzzo) | 21 | −68.88 | 18 |
| Italy (Veneto) | 21 | −91.07 | 14 |

The hailstorm over Naples being observed within a short time distance (Δt = 13 min) shows a ΔTB = −22.72 K with respect to the other two cases where very large TB discrepancies between GMI and MHS are calculated. Therefore, even though all case studies consistently respect the general 30 min criterion, the condition of simultaneity is satisfied only by the event over Naples. This supports our decision to use the hailstorm over Naples for defining the satellite pair MHS-GMI (see Table 3).

However, the hailstorms over Abruzzo and Veneto are instrumental to roughly quantify the net effect of microphysics variability on the satellite measurement: while the time distance and the MHS scan position are almost the same, a very large discrepancy between the two ΔTB values is observed (ΔTB = −68.88 and −91.07 K, respectively). These values can be justified on the basis of the system dynamics, which govern the two hailstorms and determine the different response to the satellite. Thus, by considering the hailstorms over Abruzzo and Veneto as two simultaneous events (Δt = 21 min) resolved at the same spatial scale (MHS scan position 18 and 14, respectively), the ΔTB values can be ideally conceived as the measurement of the signal attenuation due to the different cloud microphysical structures where frozen hydrometeors extinguish the upwelling radiation to the satellite.

The most important conclusions of this discussion around the three key variables are (1) no large discrepancy due to the different sampling frequency is expected; (2) the spatial resolution impacts more than the scanning mechanism in describing the ΔTBs; (3) the simultaneity of measurements plays the most important role in explaining the TB discrepancies between GMI and MHS; and (4) the microphysics modifies the signal to the satellite by characterizing each event.

*2.1. Adjustment of the GPM-C on the MHS*

As discussed above, the aim of this work is the adjustment of the GPM-C channels in the 150–170 GHz range on the MHS channel at 157 GHz in order to directly apply the MWCC-H model.

To facilitate the matching of each pair identified in Table 3, the entire database was remapped on a common $16 \times 16$ km$^2$ rectangular grid to be consistent with the MHS spatial resolution at nadir. Five sensitivity classes were defined around the absolute minimum of TBs measured by each sensor then incremented of 20 K as follows:

$$cl_i = [(i-1)\cdot20 + TB_{min} \; ; \; i\cdot20 + TB_{min}] \quad i = 1\dots5. \tag{1}$$

The definition of a neighborhood around the absolute minimum is justified by the physics behind the investigation. This analysis is focused on quantifying the divergence from the reference instrument

(MHS) of each MHS-like sensor by exploring the hail system from the maximum to the minimum scattering regions. Our results have demonstrated that the maximum divergence occurs when strong scattering by large ice is detected. In this case, the high spatial resolution and the higher sounding frequency of all MHS-like radiometers (see Table 2) with respect to MHS tend to enhance the TB depression when hail clouds are observed [26]. Hail is generally clustered in a portion of the storm cloud and the extinction by hail scattering is mainly due to such portions. Thus, instruments equipped with high frequency channels at high spatial resolution tend to measure stronger attenuation of the radiation field with respect to the instruments with worse performances. However, as concluded in the previous subsection, no large displacements are expected when frequency channels in the 150–170 GHz range are compared with MHS channel at 157 GHz. Nevertheless, large TB deviations are observed when MHS is matched with GMI due to the higher spatial resolution of the latter. Therefore, by using the five sensitivity classes built around the TB absolute minima of each satellite pair, we can quantify the signal divergence from the first class ($cl_1$, lowest TB values) to the fifth class ($cl_5$, highest TB values). The sectioning of the TBs was extremely useful to derive the predictor coefficients to adjust the MHS-like channels in the 150–170 GHz range in order to fit the hail detection model originally developed for MHS (see Equation (4) in [20]).

The adjustment procedure was formally based on a linear regression where each MHS-like channel in the 150–170 GHz frequency range is adjusted on the MHS frequency at 157 GHz ($TB_{MHS}^{157}$). A new set of TBs ($TB_i^{157}$) regressed by the minimum TB values for each class is calculated as follows:

$$TB_i^{157} = a_i TB_{MHS}^{157} + b_i \tag{2}$$

where $i$ identifies the MHS-like radiometer. Being the regressed TBs for MHS-like sensors ($TB_i^{157}$) correlated with the original data ($TB_i$) within 0.99 K, we can assume $TB_i^{157} \approx TB_i$ for deriving the MHS reconstructed signal at 157 GHz ($TB_{MHS-i}^R$). The minimization of errors was also applied to Equation (1) to calculate a set of TB-based correction terms useful to provide the optimal convergence of data. The explicit form of the final equation is, then

$$TB_{MHS-i}^R = \frac{TB_i - b_i}{a_i} + C_i^{th} \tag{3}$$

where $C_i^{th}$ designates the correction terms calculated on the basis of the TB threshold (*th*) values for each pair of radiometers. The linear regression described by Equation (3) is applied to all MHS-like sensors for reconstructing the MHS signal at 157 GHz ($TB_{MHS-i}^R$) that will feed the hail detection model MWCC-H.

### 2.1.1. MHS vs. ATMS

The ATMS sensor onboard the Suomi-National Polar-Orbiting Partnership (Suomi-NPP) and NOAA-20 is the successor of the AMSU radiometer [27]. The 22 frequency channels cover the frequency range of the Advanced Microwave Sounding Unit A (AMSU-A) and AMSU-B/MHS together with two more channels in the water vapor absorption band at 183.31 GHz (183.31 ± 4.5 and 183.31 ± 1.8). As demonstrated in Section 2, where the MHS-ATMS pair was used to evaluate the effect of the different sampling frequency, no strong effects are expected when ATMS fits the MWCC-H model. In order to adjust the ATMS channel at 165.5 GHz on the MHS channel at 157 GHz we select a severe hailstorm that hit Sicily on 22 January 2015. This rather out-of-season storm was reinforced by a heavy transport of Saharan dust triggering the formation of large hail (1.0–4.0 cm) battering first western Sicily and then the Ionian coast of Basilicata during the evening.

Although we found only one case, this event is extremely useful for reconstructing the signal at 157 GHz ($TB_{MHS-ATMS}^R$). The simultaneity of observations ($\Delta t$ = 8 min) and the similar spatial resolution (ATMS FOV 93; MHS FOV 79) created an almost perfect match between the two sensors for a better evaluation of the signal discrepancy. In Figure 2, the signal at 165.5 GHz is compared with that

at 157 GHz. The good matching between the two images identifies the main core of storm where the maximum signal reduction is registered. The TB values along a transect crossing the convection core show the enhancement of the signal attenuation near the core where values of 154.31 K and 153.73 K are measured from ATMS and MHS, respectively. The small difference between the two measurements suggests that ATMS can be directly used in the MWCC-H model, but the calculation of the regression coefficients through Equation (3) is essential for a fine-tuning of the two sensors.

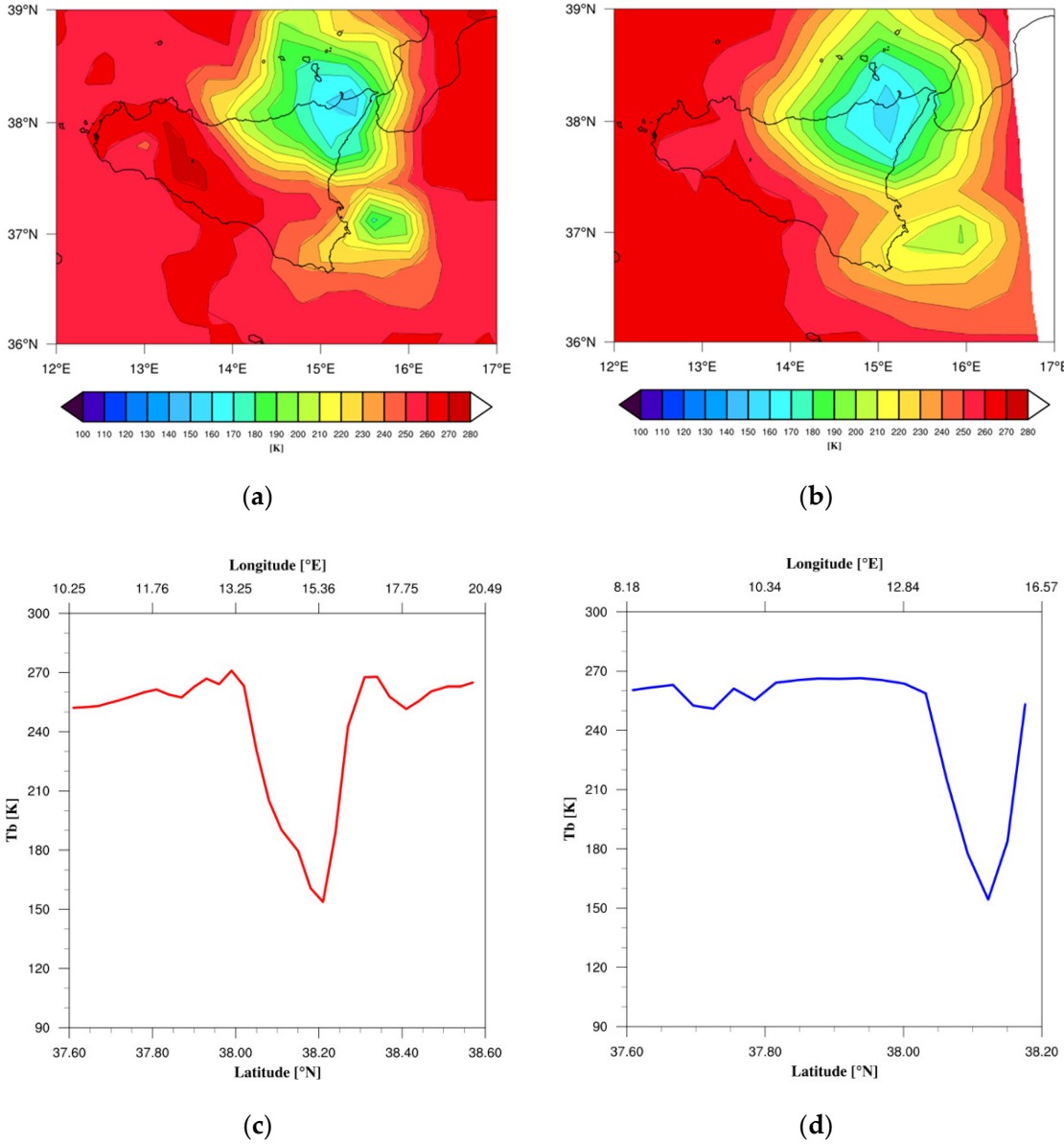

**Figure 2.** Brightness temperature (TB) distribution as measured from the N19-MHS at 13:04 UTC (**a**) and NPP-ATMS at 12:56 UTC (**b**) during the severe hailstorm over Sicily (Italy) on 22 January 2015. Diagrams (**c**) and (**d**) show the corresponding distributions of TBs along a transect through the core of the storm system reaching almost the same minimum values where the hail core is observed.

By sequencing all TBs into the five classes the effect of different sounding frequencies can be evaluated and the regression predictors of Equation (3) calculated. Since the five classes are conceived as a sort of "tracker" of signal variation from hail to "no-hail" regions, the minimum TB value is a good indicator of the scattering intensity from the hydrometeors that reduces the radiation field. Table 5 shows the minimum TB value for each class. The difference $\Delta_{ATMS}$ oscillates around zero in the first

two classes indicating a high overlapping where the main core is located. Values found in the last three classes slightly diverge from zero because of the changing cloud microphysics that smoots the effect of scattering from the hydrometeors, which progressively change phase and size.

**Table 5.** TB minimum values for each class for MHS and ATMS. Italics highlights the signal difference $\Delta_{ATMS}$ ($TB_{ATMS} - TB_{MHS}$) between the two instruments when calculated for each sensitivity class.

| SatID | Class 1 | Class 2 | Class 3 | Class 4 | Class 5 |
|---|---|---|---|---|---|
| MHS | 153.73 | 177.20 | 197.85 | 216.11 | 233.78 |
| ATMS | 154.31 | 176.58 | 194.47 | 214.65 | 234.42 |
| $\Delta_{ATMS}$ | *0.58* | *−0.62* | *−3.38* | *−1.46* | *0.64* |

Note: Italics is used just to better highlights the difference of signal when the two sensors observe the same region of the storm.

Therefore, by applying the adjustment equation for the ATMS channel at 165.5 GHz (Equation (2)) we can reconstruct the MHS channel at 157 GHz through Equation (3) and calculate the TB-based correction terms (Table 6) by a convergence method:

$$TB^{157}_{MHS-ATMS} = \frac{TB_{ATMS} + 3.9855}{1.0178} + C^{th}_{ATMS} \tag{4}$$

**Table 6.** Correction terms for ATMS as a function of TB thresholds.

| $C^{th}_{ATMS}$ | TB (K) |
|---|---|
| 1.7449 | >234 |
| −1.0071 | 216 ÷ 234 |
| −0.8656 | 195 ÷ 216 |
| 0.2584 | 174 ÷ 195 |
| −3.4761 | <174 |

The application of the MWCC-H hail model fed by the MHS reconstructed signal at 157 GHz ($TB^R_{MHS-ATMS}$) shows the high performance of the model in detecting the hail system as demonstrated by the good match with the hail probability derived from the MHS original data. Both retrievals identify a probability value in the 0.50–0.60 range, which is typically associated with large hail with diameters between 2 to 10 cm. The few discrepancies between the two images in Figure 3a,b are justified by the slightly higher resolution of the MHS FOVs with respect to that of the ATMS positioned to the edge of scanline.

Finally, the temperature–hail (T-H) proxy curve in Figure 3c correlates the adjusted ATMS channel at 165.5 GHz through Equation (3) with the probability associated with hail detection. The T-H diagram initially proposed by Laviola et al. [20] (see their Figure 8b) is proposed as a general slide rule to quickly identify the hail diameter category by crossing the TB values with the hail probability range as described by the authors and here replicated in Table 7.

**Table 7.** Probability range for identifying hail category.

| Probability of Hail | Diameter Range (cm) | Category Description |
|---|---|---|
| 0.36 ÷ 0.45 | <2 | Graupels/Hail Initiation (HI) |
| 0.45 ÷ 0.60 | 2 ÷ 10 | Large Hail (H) |
| >0.60 | >10 | Super Hail (SH) |

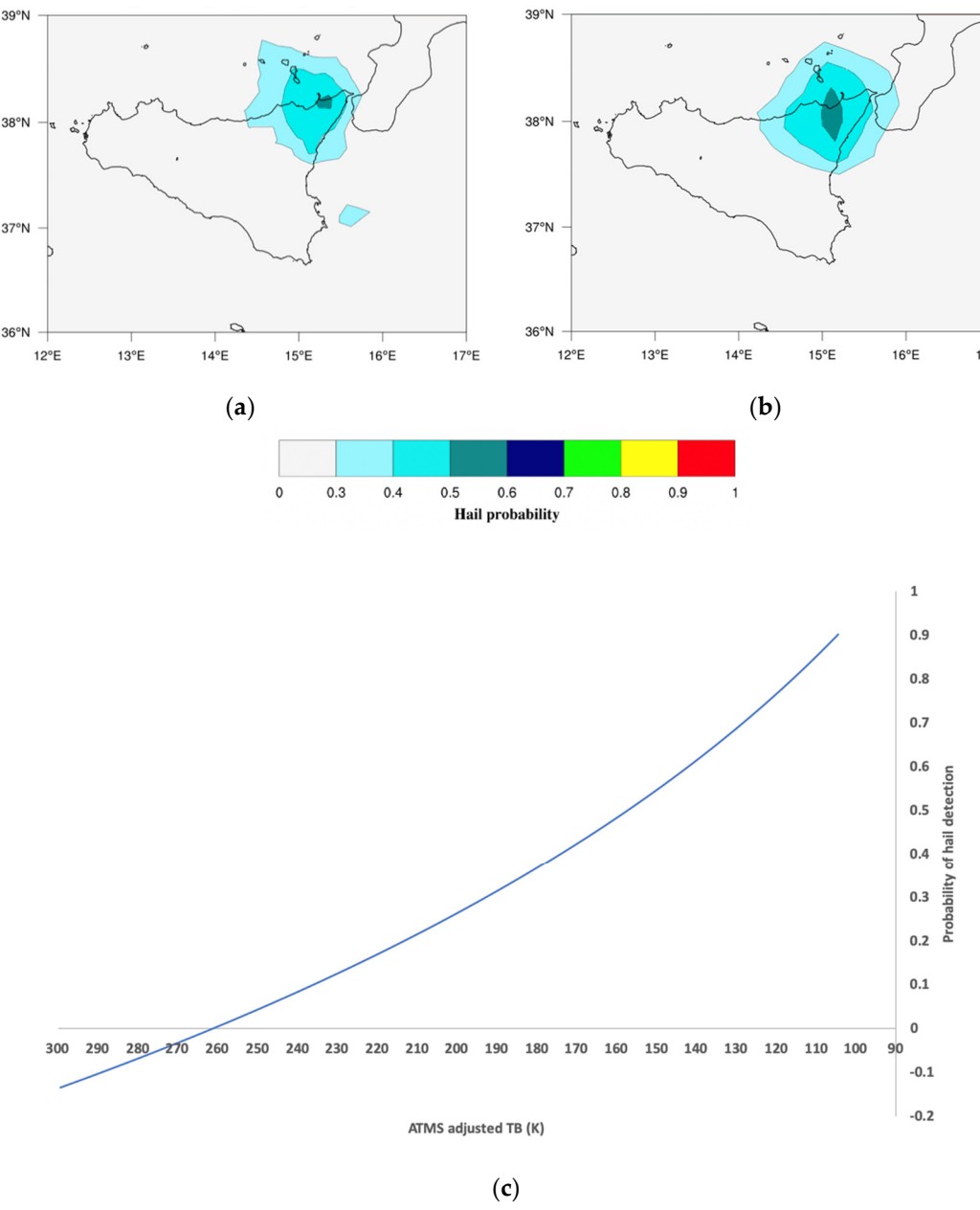

**Figure 3.** Same as in Figure 2, but for hail probability calculated by the MWCC-H model for MHS (**a**) and ATMS (**b**). Although the morphology of the hail cloud is slightly different, the distribution of hail probability identifies the same regions of deep convection. The diagram in (**c**) is the T-H proxy curve for ATMS useful for detecting the hail categories (see Table 7).

### 2.1.2. MHS vs. SSMIS

The SSMIS is a 24-channel radiometer flying aboard the US DMSP F-16, F-17, F-18, and F-19 satellites. Being the successor of the SSM/I, the SSMIS combines imaging and sounding capabilities covering the wide frequency range from 19 GHz to 183 GHz with 21 linearly polarized frequency channels [28]. The SSMIS conical scanning has been initially used as an indicator for evaluating the impact of different scanning mechanisms when compared with MHS. In the second step, the SSMIS 150 GHz horizontally polarized channel is used for reconstructing the MHS channel at 157 GHz to feed the MWCC-H model. The severe hailstorm over North Dakota on 10 June 2017 was selected for evaluating the effect of the different scanning mechanisms and calculating the reconstructed signal $TB^R_{MHS-SSMIS}$.

Through a pixel-by-pixel analysis of the hailstorm, we found a similar signal distribution, but while approaching the storm core, a divergence of about 20 K between the minimum values is observed. However, as discussed in Section 2, a low confidence is attributed to the association of this TB displacement to the "pure" impact of the different scan mechanism of the MHS-SSMIS pair because of the large difference of the spatial resolution between the two sensors (MHS FOV 66).

In Figure 4a portion of the hailstorm with minimum TB values observed by the SSMIS is compared with the corresponding MHS observation. The notable agreement in distinguishing the main three cores of the storm is contrasted by several cold regions observed by the SSMIS where the majority of large hail is distributed. Furthermore, the better resolution of SSMIS allows a more detailed observation of the storm cores appearing larger than that observed from the MHS. The corresponding diagrams gathering the signal from the hail core 2 (blue arrow in Figure 4b) quantify the differences of the two instruments by displaying a cuspidal distribution of TBs measured from the MHS (Figure 4a) with respect to the wide cold region around the minimum as seen by the SSMIS.

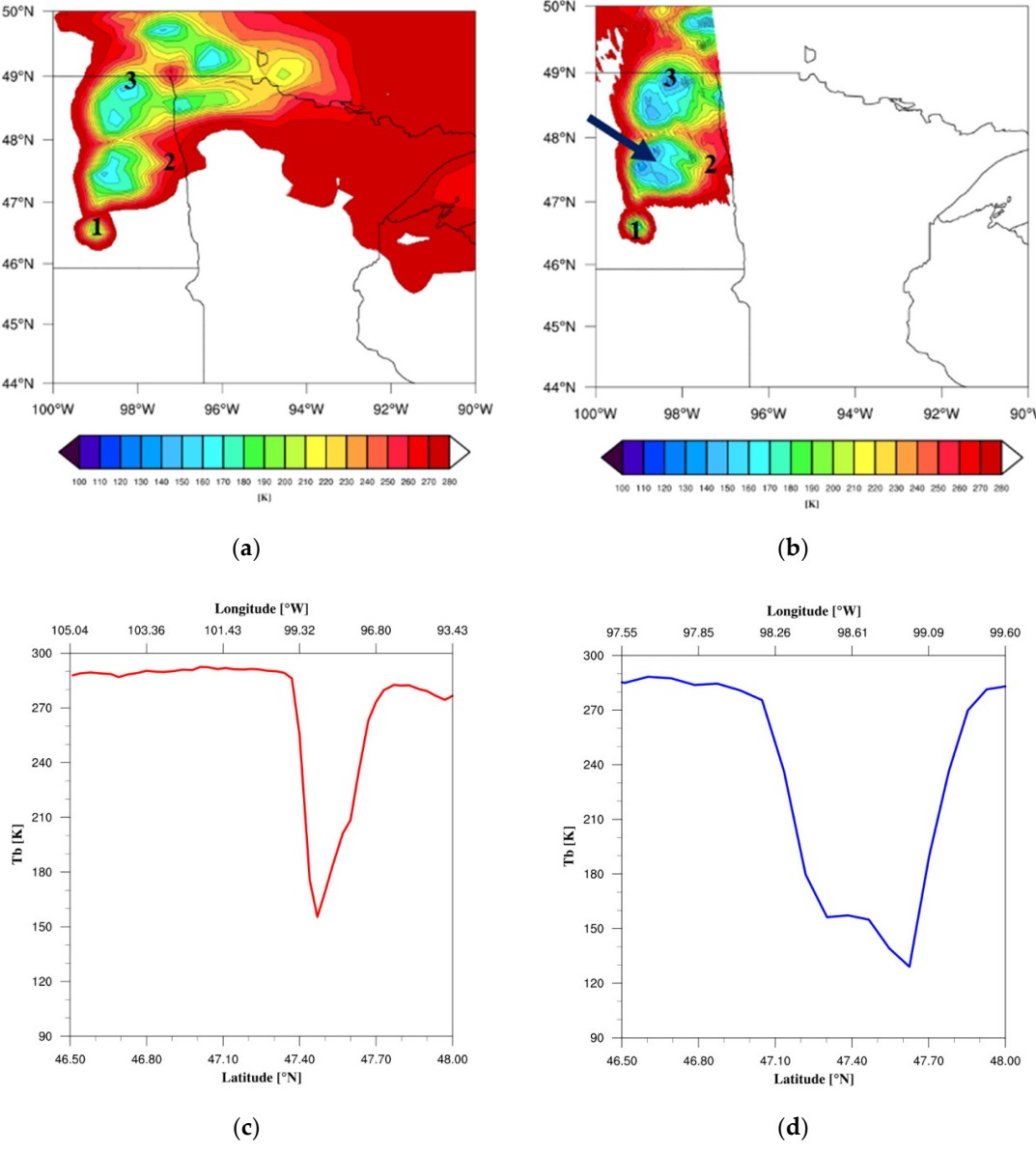

**Figure 4.** TB distribution as measured from the N18-MHS at 0110 UTC (**a**) and F17-SSMIS at 01:02 UTC (**b**) during the severe hailstorm over North Dakota on 10 June 2017. The lower diagrams (**c**,**d**) show the distribution of TBs along the transect through core 2 (see blue arrow) reaching the absolute minimum.

The application of the classes built around the minima at 155.42 K and 135.21 K for MHS and SSIMS, respectively, identifies a progression of relatively high TB values mirroring not very large hail diameters (Table 8). In fact, on the basis of the evaluations described in Laviola et al. [20], MHS TBs higher than 152.51 K are correlated with hail diameters in the 2–10 cm range.

**Table 8.** TB minimum values for each class for MHS and SSMIS. Italics highlights the signal difference $\Delta_{SSMIS}$ ($TB_{SSMIS} - TB_{MHS}$) between the two instruments when calculated for each sensitivity class.

| SatID | Class 1 | Class 2 | Class 3 | Class 4 | Class 5 |
|---|---|---|---|---|---|
| MHS | 155.42 | 175.44 | 197.77 | 215.71 | 235.53 |
| SSMIS | 135.21 | 155.23 | 175.47 | 195.85 | 215.67 |
| *$\Delta_{SSMIS}$* | *−20.21* | *−20.22* | *−22.30* | *−19.86* | *−19.86* |

Note: Italics is used just to better highlights the difference of signal when the two sensors observe the same region of the storm.

As noted from Table 8, the differences $\Delta_{SSMIS}$ show values regularly distributed around 20 K, giving an intrinsic reproducibility to the distribution.

Thus, using the adjusted SSMIS signal at 150 GHz (Equation (1)), Equation (3) can be specialized for calculating the reconstructed MHS TBs at 157 GHz, as follows:

$$TB_{MHS-SSMIS}^{157} = \frac{TB_{SSMIS} + 49.923}{1.1530} + C_{SSMIS}^{th} \tag{5}$$

A convergence method is then used to calculate the TB-based correction terms listed in Table 9.

**Table 9.** Tabulation of correction terms for SSMIS as a function of TB thresholds.

| $C_{SSIMS}^{th}$ | TB (K) |
|---|---|
| −5.3656 | >240 |
| 3.6134 | 195 ÷ 240 |
| 1.6910 | 180 ÷ 195 |
| −0.4578 | 154 ÷ 180 |
| −7.3520 | <154 |

In Figure 5 the detection of hail clouds is done from the MHS reconstructed signal at 157 GHz ($TB_{MHS-SSMIS}^{R}$) used for feeding the MWCC-H model. At first sight, we note that the higher spatial resolution of the SSMIS identifies more hail cores than the reference instrument. This is expected because Equation (3) performs only a radiometric adjustment of the targeted frequency channels; then the geometric characteristics follow the native instrumental configuration of the MHS-like radiometers.

However, a more detailed inspection of the results demonstrates the good matching of the two detections where the structure of the hail clouds are almost the same and the small nested hail areas are morphologically and quantitatively similar. The innermost core where the extinction of radiation due to scattering is maximum (see blue arrow in Figure 4b) is resolved by both instruments and classified in the 0.60–0.70 range, which is generally associated with the transition between large hail and super hail. The rest of the storm is categorized in the 0.40–0.50 and 0.30–0.40 ranges from the center to the cloud edge, respectively. A useful interpretation key of the SSMIS performances in reconstructing the hail field is the T-H proxy curve displayed in Figure 5c correlating the adjusted SSMIS TBs on the basis of Equation (3) with the probability of hail.

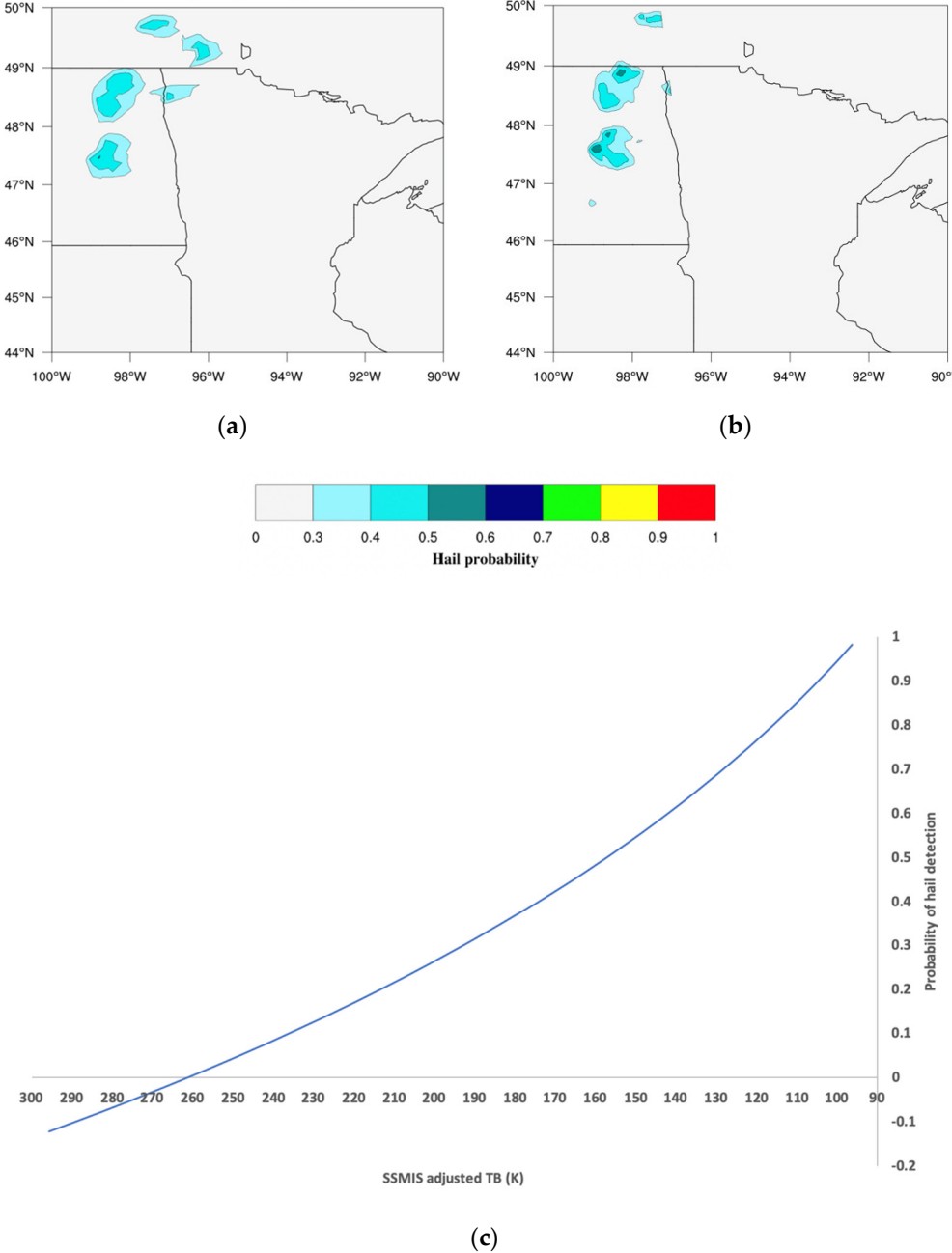

**Figure 5.** As in Figure 4 but for hail detection from MHS (**a**) and SSMIS (**b**). Although the morphology of hail cloud is slightly different the distribution of hail probability delineates the same regions of deep convection. The diagram in (**c**) is the temperature–hail (T-H) proxy curve for SSMIS useful for detecting the hail categories (see Table 7).

### 2.1.3. MHS vs. GMI

The GMI is a multichannel conical-scanning radiometer characterized by 13 frequency channels covering the range from 10.65 GHz to 183.31 GHz. The radiometric sensitivity and the possibility offered by the simultaneous measurements with the DPR make these sensors the reference of the GPM Constellation. In this section the potential of the passive and active sensors of the GPM-CO are used to better investigate the selected hailstorm events. The DPR measurements when co-located with the GMI provide more complete information in understanding the extinction of the radiation field due to hail. The sounding of the vertical distribution of hydrometeors and the measurement of their direct impact on the radar reflectivity offer a robust tool for exploring the TB reduction related to hail scattering.

Figure 6 displays the radiation fields measured by the MHS and GMI frequency channels at 157 and 166.5V GHz, respectively, during the severe hailstorm that hit the Gulf of Naples in Southern Italy on 5 September 2015. As mentioned above, the choice of this event for adjusting the GMI channel at 166.5V GHz on the MHS channel at 157 GHz was driven by the nearly simultaneous measurements MHS-GMI ($\Delta t$ = 13 min) by means of the measured TB discrepancies ($\Delta$TB = $-$22.72 K) due to the large difference in spatial resolution.

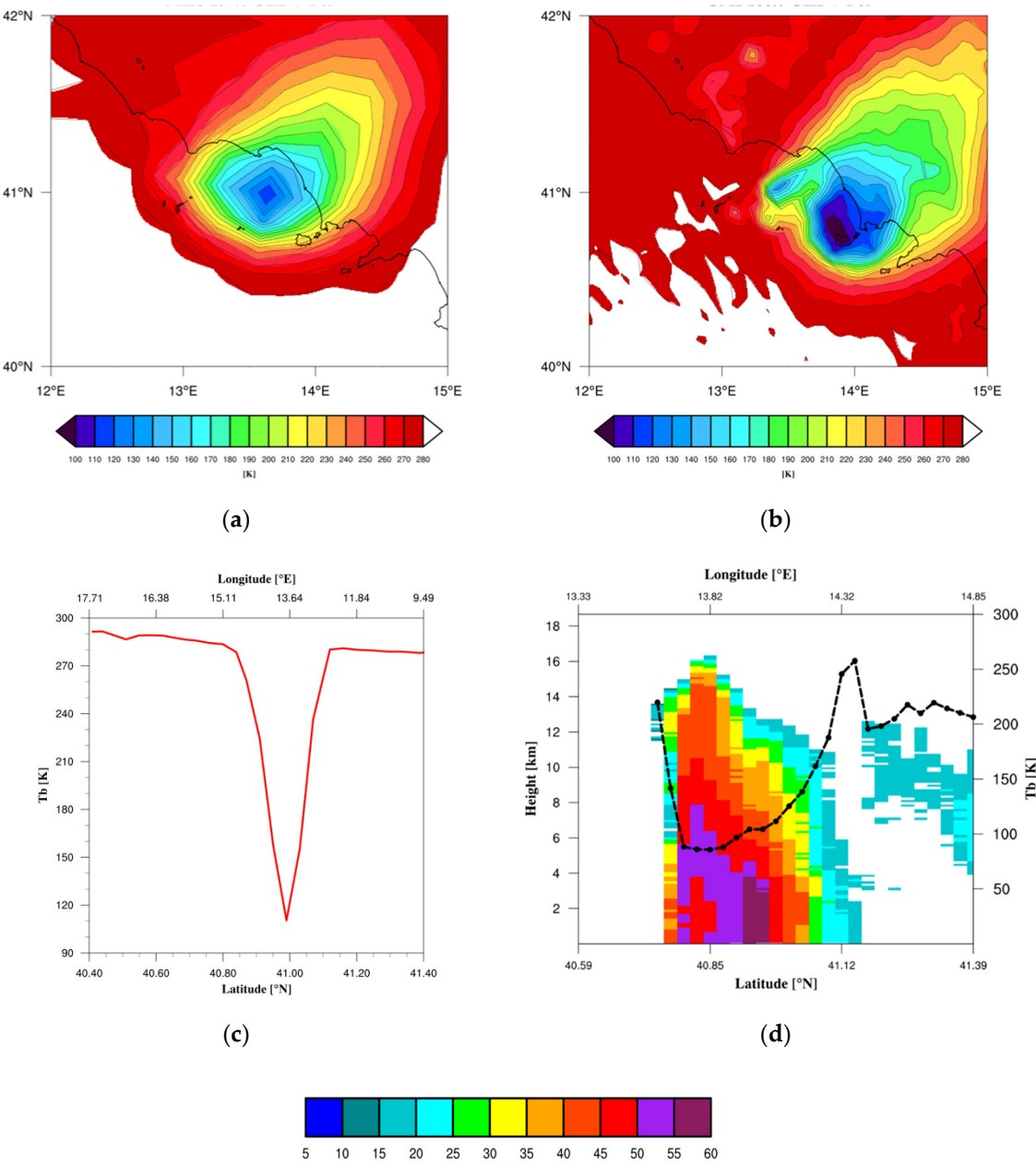

**Figure 6.** TB distribution as measured from the MO2-MHS at 08:34 UTC (**a**) and GPM-GMI at 08:47 UTC (**b**) during the severe hailstorm over Naples (Italy) on 5 September 2015. The corresponding TB distributions along a transect through the storm core (**c,d**) show the TB reduction reaching the maximum value on the main hail core of the storm. Further information is provided by the DPR Ku-band measurement offering a vertical cross section of the hail core with maximum reflectivity values.

The deviation of two radiation fields is quantified by the diagrams in Figure 6, where a V-shaped distribution of the MHS TBs matches with a well-defined trend of signal reduction measured by the

GMI. The coarse resolution of the MHS generally limits the observation of small-scale phenomena typically arising on a scale of the order of kilometers like hailstorms or hail cores embedded into large convections. Hence, GMI better resolves the leading features of such local events by showing the progressive attenuation of the radiation field approaching the core of the storm. The co-located Ku-band contributes to define the inner structure of convection by highlighting the most active part of the system (50–60 dBZ) which extinguishes the radiation where a large hail volume is located.

The classification of TB values around the minima at 110.73 K and 96.44 K for MHS and GMI, respectively, reveals the severity of the hailstorm characterized by exceptional intensity and rare rapid development. The low values found from the five classes (Table 10) identify hail with very large diameters (>10 cm) as documented by the official reports of the ESWD (see Table 3), typically associated with the super hail category [20]. The differences of the minima ($\Delta_{GMI}$) exhibit large discrepancies in all classes indicating that the high resolution of GMI significantly controls the matching with MHS. The latter consideration is supported by looking at $\Delta_{GMI}$ values distributed for each class: the observed opposite magnitudes of the first classes ($cl_1$, $cl_2$) where the scattering intensity is normally very high, with respect to the last two ($cl_4$, $cl_5$), indicate that small-scale cloud structures detected by GMI affect the signal reduction more than those observed by MHS where the worse spatial resolution smooths these effects.

**Table 10.** TB minimum values for each class for MHS and GMI. Italics highlights the signal difference $\Delta_{GMI}$ ($TB_{GMI} - TB_{MHS}$) between the two instruments when calculated for each sensitivity class.

| SatID | Class 1 | Class 2 | Class 3 | Class 4 | Class 5 |
|---|---|---|---|---|---|
| MHS | 110.73 | 134.64 | 155.54 | 179.12 | 196.60 |
| GMI | 96.44 | 116.92 | 148.18 | 156.80 | 177.81 |
| *$\Delta_{GMI}$* | *−14.29* | *−17.72* | *−7.36* | *−22.32* | *−18.79* |

Note: Italics is used just to better highlights the difference of signal when the two sensors observe the same region of the storm.

The following equation is derived from Equation (3) when applying the GMI frequency at 166.5 GHz for reconstructing the radiometric signal of the MHS channel at 157 GHz

$$TB_{MHS-GMI}^{157} = \frac{TB_{GMI} + 3.2588}{0.9612} + C_{GMI}^{th} \tag{6}$$

where the correction terms are to be found in Table 11.

**Table 11.** Tabulation of correction terms for GMI as a function of TB thresholds.

| $C_{GMI}^{th}$ | TB (K) |
|---|---|
| −0.9975 | >188 |
| 5.4268 | 163 ÷ 188 |
| −3.8086 | 149 ÷ 163 |
| 4.3922 | 122 ÷ 149 |
| −3.9719 | <122 |

Figure 7 shows the application of the MWCC-H model fed with the reconstructed signal at 157 GHz ($TB_{MHS-GMI}^{R}$) compared to the model referring to the reference instrument. The high similarity of two images in identifying the main core of the storm and distinguishing the more active regions of the system is contrasted by superior details captured by the GMI. The higher spatial resolution of the GMI allows a fine-scale reconstruction of the distribution of probability values from the core to the borders of the hail system. In spite of these expected differences between the radiometers, the two retrievals quantitatively reveal very few divergences by classifying the detected hail as super hail while fitting the surface observations. In fact, the probability values calculated in the core of the convection

are 0.86 and 0.95 for MHS and GMI, respectively. A useful tool for describing the performances of the MWCC-H fed with the adjusted GMI TBs is made available by the T-H proxy curve in Figure 7c.

Finally, note that the discrepancies of two detections can also be due to the rapid evolution of the storm that in just a few 10s of minutes completely changed its cloud structure and microstructure by changing in turn the response to the satellite.

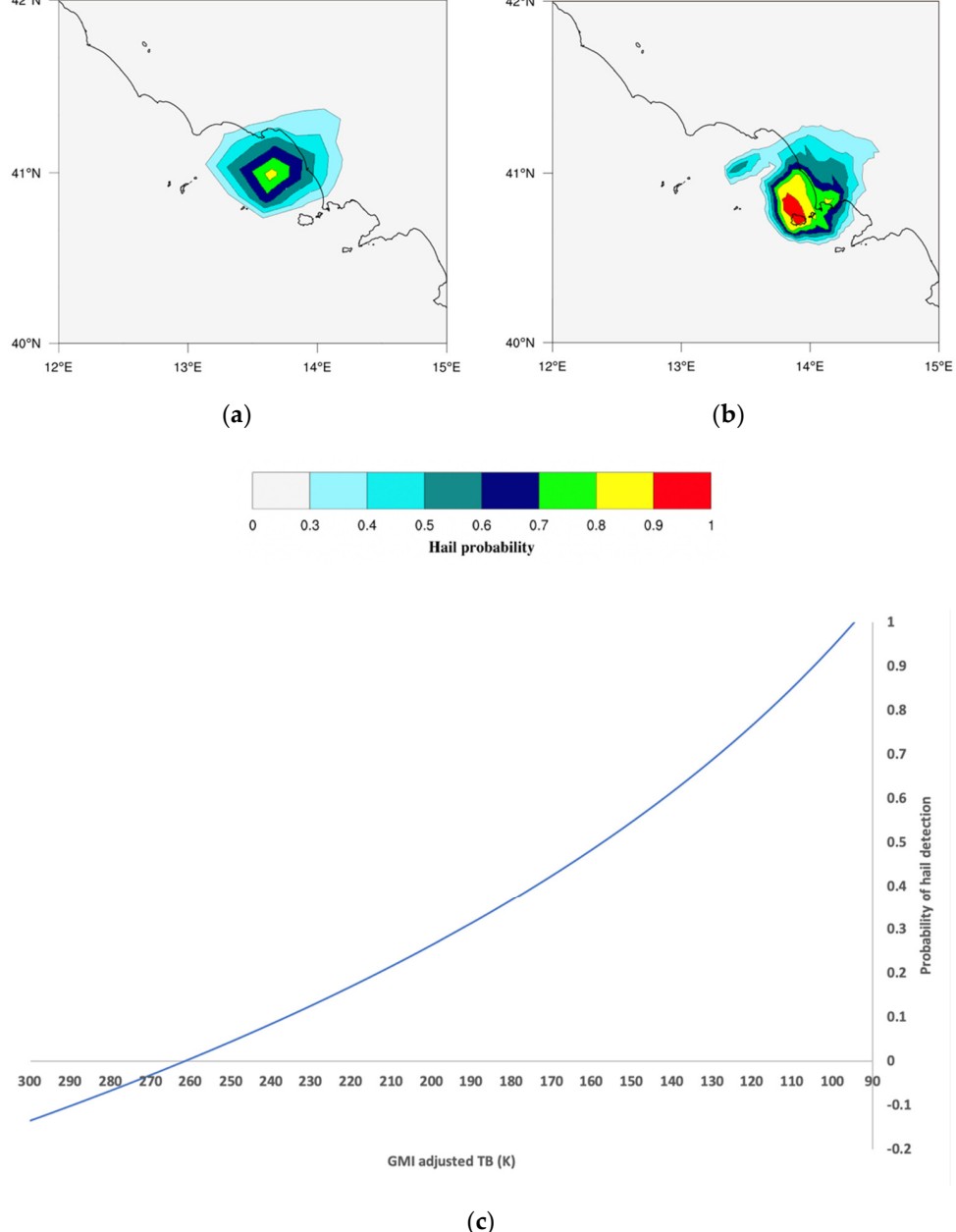

**Figure 7.** Same as in Figure 6, but showing hail detection (**a**,**b**). The diagram in (**c**) is the T-H proxy curve for GMI for detecting hail categories (see Table 11).

## 3. Real-Time Applications

In this section, the MWCC-H is applied to the GPM-CO in order to investigate the operational potential of the generalized hail detection model in describing the main features of hailstorms in terms of system morphology and extension, and intensity and distribution of hail cores.

Since no coincidence between the GPM-CO and the rest of MHS-like sensors was found, the MWCC-H has been evaluated through two separate streams.



First, the performance of the MWCC-H applied to the GPM-CO is described using two events where the GMI hail detection is coupled with the DPR Ku-band measurement of the signal attenuation due to the vertical distribution of the hail bulk. The DPR observation is extremely useful to identify the cloud regions subject to highest attenuation where the reduction of TBs and probability of hail reach the maximum values. Figure 8a,b shows a series of thunderstorms organized as a squall-line that produced a sequence of intense hailfalls along the Adriatic Sea. The evolution of the MWCC-H hail probability identifies the most active hail clouds just off the Italian coastline. Specifically, the largest hail diameters were registered in the littoral city of Casalbordino (Abruzzo region, central Italy) where hailstones with diameters >7 cm were sampled at the ground (source: ESWD archive). The transect through the most intense hail core (black arrow in Figure 8a) reveals that the strong TB reduction corresponding to the maximum value of the hail probability is associated with very high reflectivities reaching their maximum at 50–55 dBZ below 4 km.

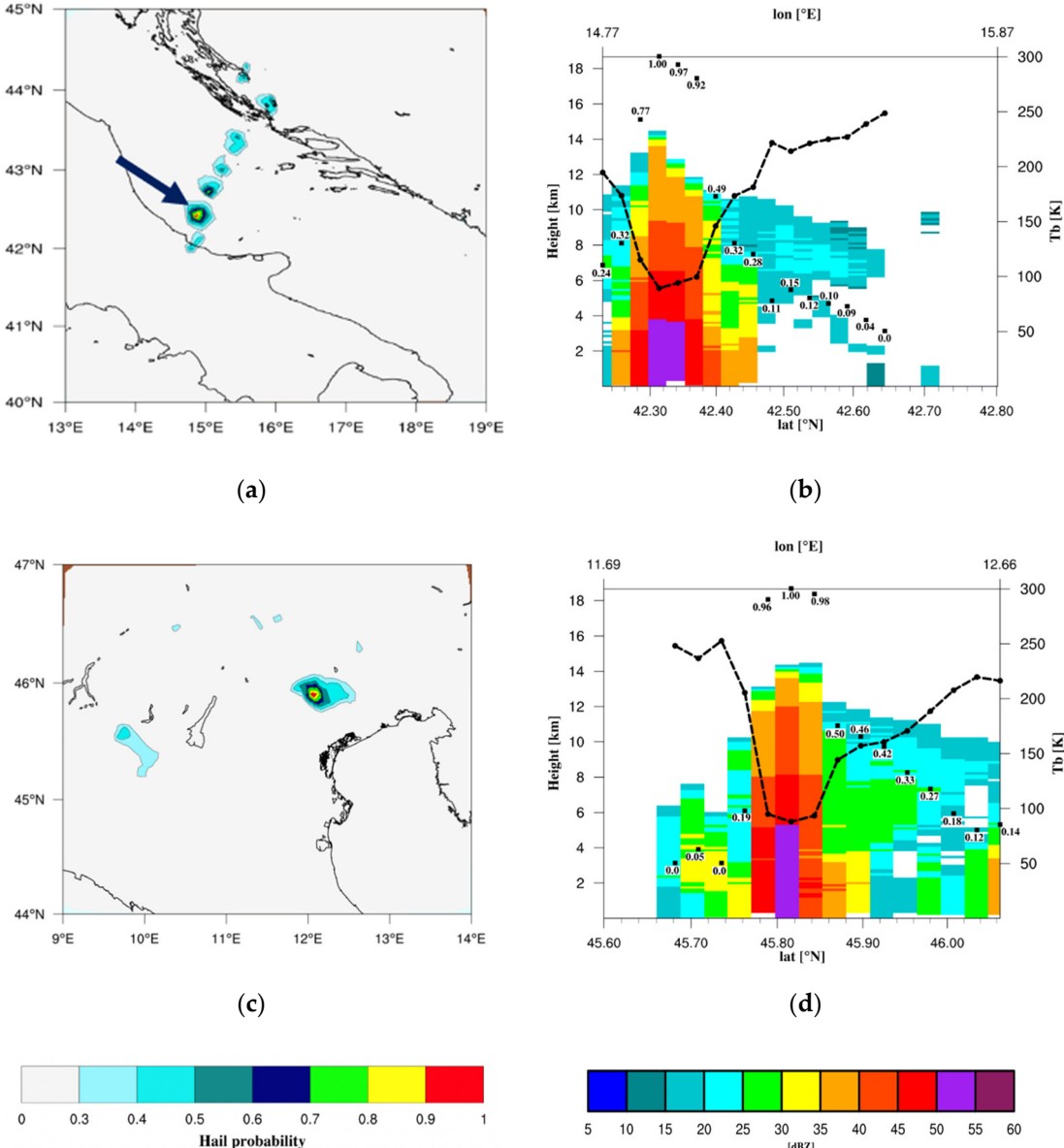

**Figure 8.** MWCC-H applied to the GPM-GMI (left column) during two hailstorm events: Adriatic Sea (Abruzzo, Italy) (**a,b**) on 15 October 2016 at 0946 UTC, Veneto region (Italy) (**c,d**) on 5 July 2018 at 1810 UTC. The distribution of hail probability corresponds to the highest reflectivity values from the GPM-DPR Ku-band (right column).

Similar considerations can be done for the other case study where an isolated convective core triggered a severe hailfall over the Veneto region (Italy; just north of Venice) on 5 July 2018. As for the previous event, the transect through the hail core reveals the deep depression of TBs reaching values lower than 100 K, which correspond to a hail probability close to 1 (see the T-H proxy curve for GMI in Figure 7). The radar reflectivity (Figure 8d) shows the vertical distribution of signal attenuation touching the maximum of 50–55 dBZ below 6 km.

The second series of applications is the chronological reconstruction of the storm trail that affected the Adriatic coast of Italy during 10 July 2019. The daily sequence starts early morning from the Venice Lagoon and propagates during the day along the Italian Peninsula hitting several coastal towns. The main characteristic of this sequence was the intensity of each event both in terms of hail size and surface wind speed. The complexity of the storm sequence, triggered by a synoptic disturbance, was enhanced by very intense local phenomena such as downbursts, waterspouts and tornadoes often associated with very large hail. Substantial damages to infrastructures and trees were documented in the cities hit by the storm.

In Figure 9, the temporally ordered sequence of hail events as detected from the whole GMP-CO (except of GMI) is shown. The first three images (Figure 9a–c) describe the early stages of the EF1 tornado hitting at 07:30 UTC the littoral city of Milano Marittima where very large hail size was associated to the storm as documented from the official report of the Regional Agency for Prevention, Environment and Energy of Emilia-Romagna (ARPAE, 2019). The next series of images (Figure 9d–f) describes the southward motion of the storm where the system drifts off-shore probably gaining vigor before triggering the most intense sequence of hailstorms lasting till the evening.

In Figure 9g,h, the MWCC-H model describes the severity of the phenomenon by capturing the hailstorm intensification stages reaching the maximum diameters of 9–14 cm on the Abruzzo coastline. The nested squared windows zoom hail clouds moving on the Adriatic littoral.

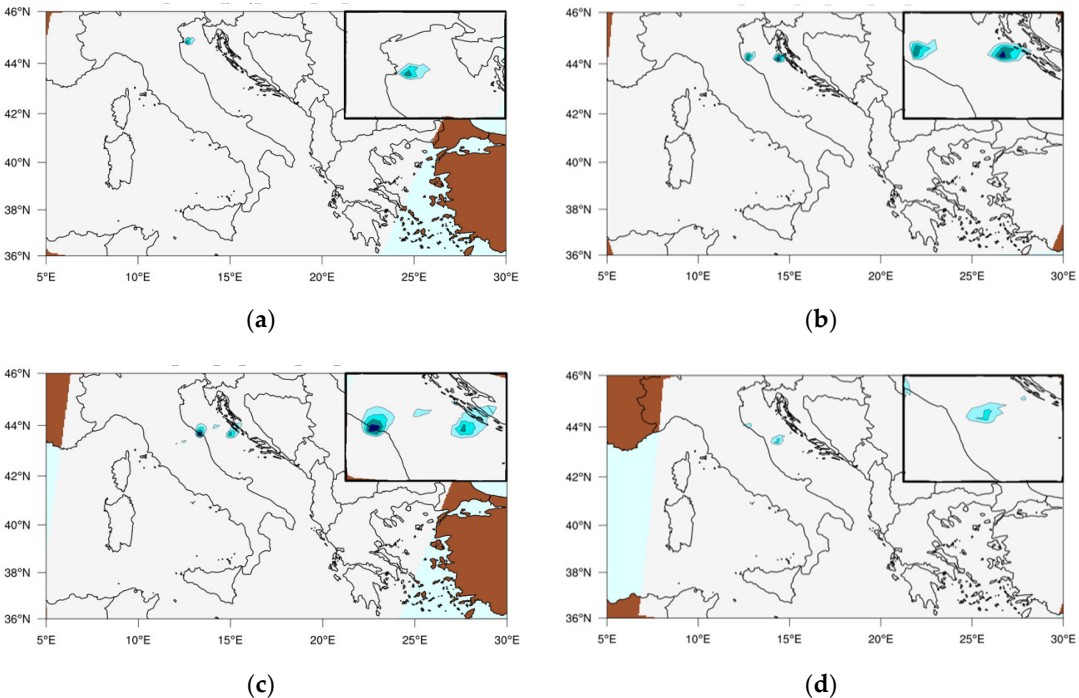

**Figure 9.** *Cont.*

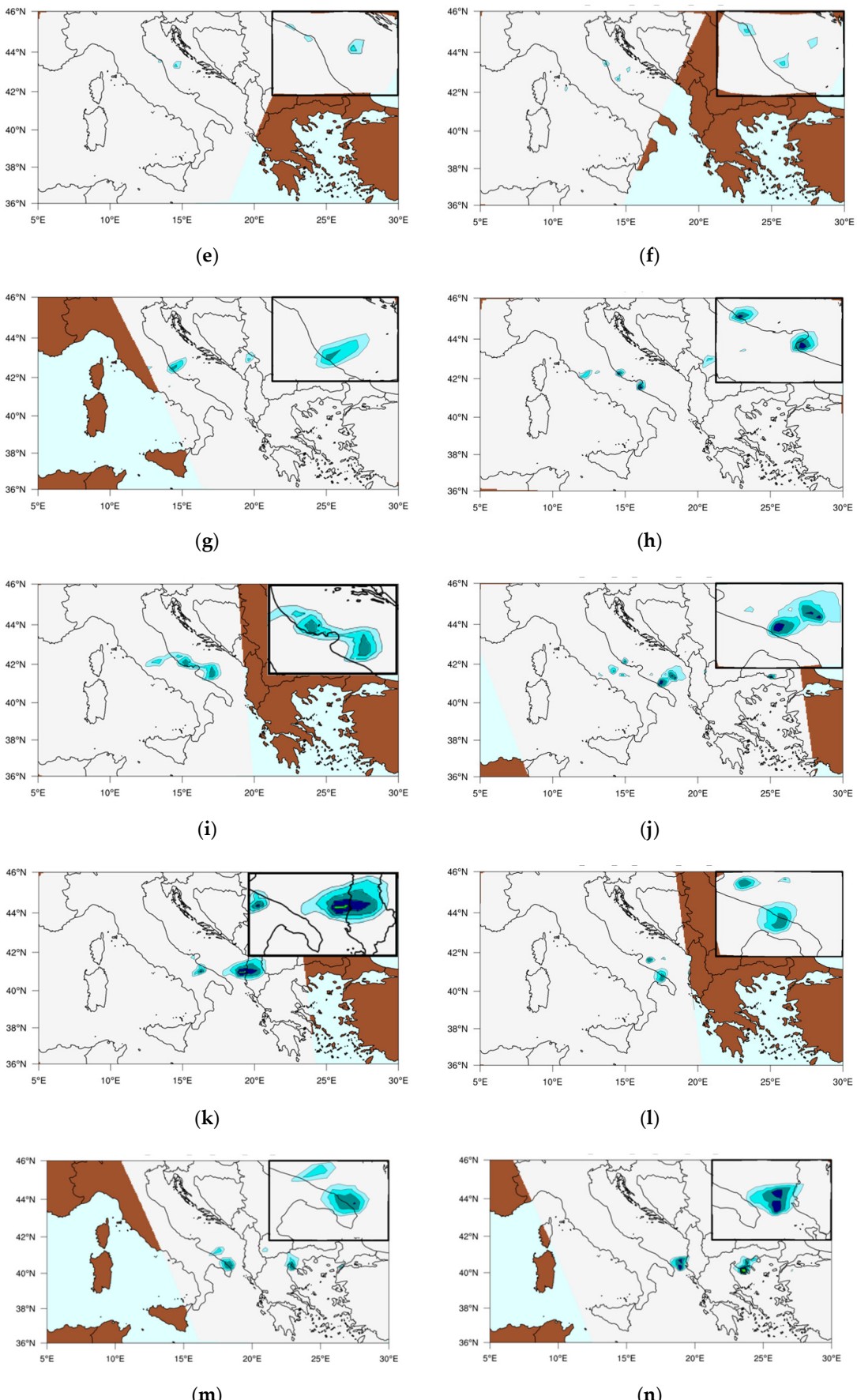

**Figure 9.** *Cont.*

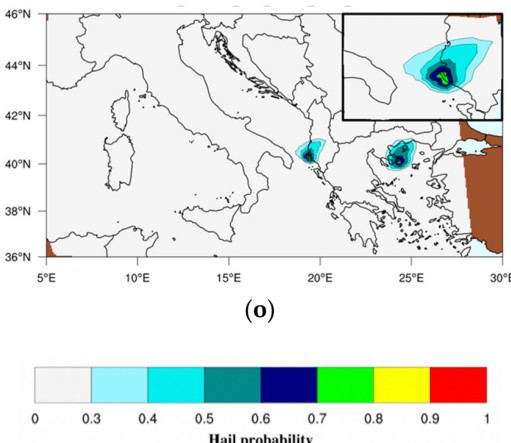

(o)

**Figure 9.** Chronological reconstruction of the daily sequence of hailstorm events affecting the Italian Adriatic regions during 10 July 2019. From (**a–o**), we see the hail probability inferred by ATMS-N20 at 01:13 UTC, MHS-N19 at 04:18 UTC, SSMIS-F17 at 06:03 UTC, MHS-MOB at 08:35 UTC, MHS-MOA at 09:01 UTC, MHS-MOC at 09:42 UTC, ATMS-N20 at 10:57 UTC, ATMS-NPP at 11:47 UTC, ATMS-N20 at 12:37 UTC, SSMIS-F16 at 14:18 UTC, MHS-N19 at 15:43 UTC, SSMIS-F17 at 17:30 UTC, MHS-MOA 18:40 UTC, MHS-MOC at 19:21 UTC, and MHS-MOB at 19:55 UTC. Squared windows zoom the most intense hail clouds affecting the Adriatic littoral.

The next series of images (Figure 9i–o) reconstructs the hail events that occurred during the afternoon of 10 July 2019. The system moved southward, and the development of hailstorms intensified with the most intense events of the day hitting southern Italy and the Balkans. Such intense hail clusters first hit the Apulia region, where the MWCC-H hail probabilities greater than 0.6 indicate the formation of clouds with super hail bands as confirmed by the ESWD archive with hail diameters up to 8 ÷ 10 cm. Later, the storm moved to Albania and Greece where the transition between large hail and super hail was often associated with waterspouts or funnel clouds.

The applications in Figure 9 show the potential of the MWCC-H model in describing the complete evolution of hailstorms. These are currently limited to polar-orbiting passive microwave observations, but the potential of such measurements demonstrates to be extremely useful if they were available from geostationary (GEO) orbit. Hence, it can be concluded that in this case the MWCC-H model realistically replicates the characteristics of a GEO sensor to continuously monitor the evolution of a thunderstorm

## 4. Hail Detection at the Global Scale

Since the MWCC-H model was originally developed and tested on hail events scattered around the world, it is now appropriate to evaluate the model performance for global applications. Therefore, applying the same 2008 AMSU-B observations used by Ferraro et al. [19], a global map of the detected hail events has been derived. The main objective of this application is to evaluate the capability of the MWCC-H model to reproduce the areal distribution of hail events during the year. Thus, drawing a seasonal map of hail events is the one of the main focus of this application.

The overall good performance of the MWCC-H model in reconstructing the annual transition of hailstorms between the two hemispheres is supported by the sequence of images proposed in Figure 10 where the AMSU-B observations for the year 2008 have been used for detecting the occurrence of hail events worldwide. As expected, the distribution of the events reveals that during the winter months the majority of countries hit by hail are located in the southern hemisphere: Latin America, South Africa and Australia are affected by the largest numbers of hailfalls in that season.

The northward shift of the hail clusters starts during March/April and reaches the maximum number of occurrences during the warm season (May/June/July) and decreases when approaching fall. Note that the maximum of occurrences for northern and southern hemispheres have been

respectively found in May/June in the US Great Plains and October/November in the South America. Further considerations can be drawn on the tropical regions where the number of detected hail events could be unrealistic. Due to the large amount of water vapor, tropical convections extend very high in the atmosphere and produce a very high concentration of frozen hydrometeors. As for the work of Ferraro et al. [19], no corrections were made on the AMSU-B data for computing the global annual climatology, thus the occurrences of hail in the tropical regions can be overestimated due to the significant ice amount in the convective clouds that only partially fall to the ground.

Finally, it should be noted that the brown areas are due to cold and/or high elevation terrains, where the algorithm cannot distinguish between frozen surface and hail. These spurious effects are reduced in the last version of the MWCC-H, where surface elevation data and a snow cover mask have been implemented in the code.

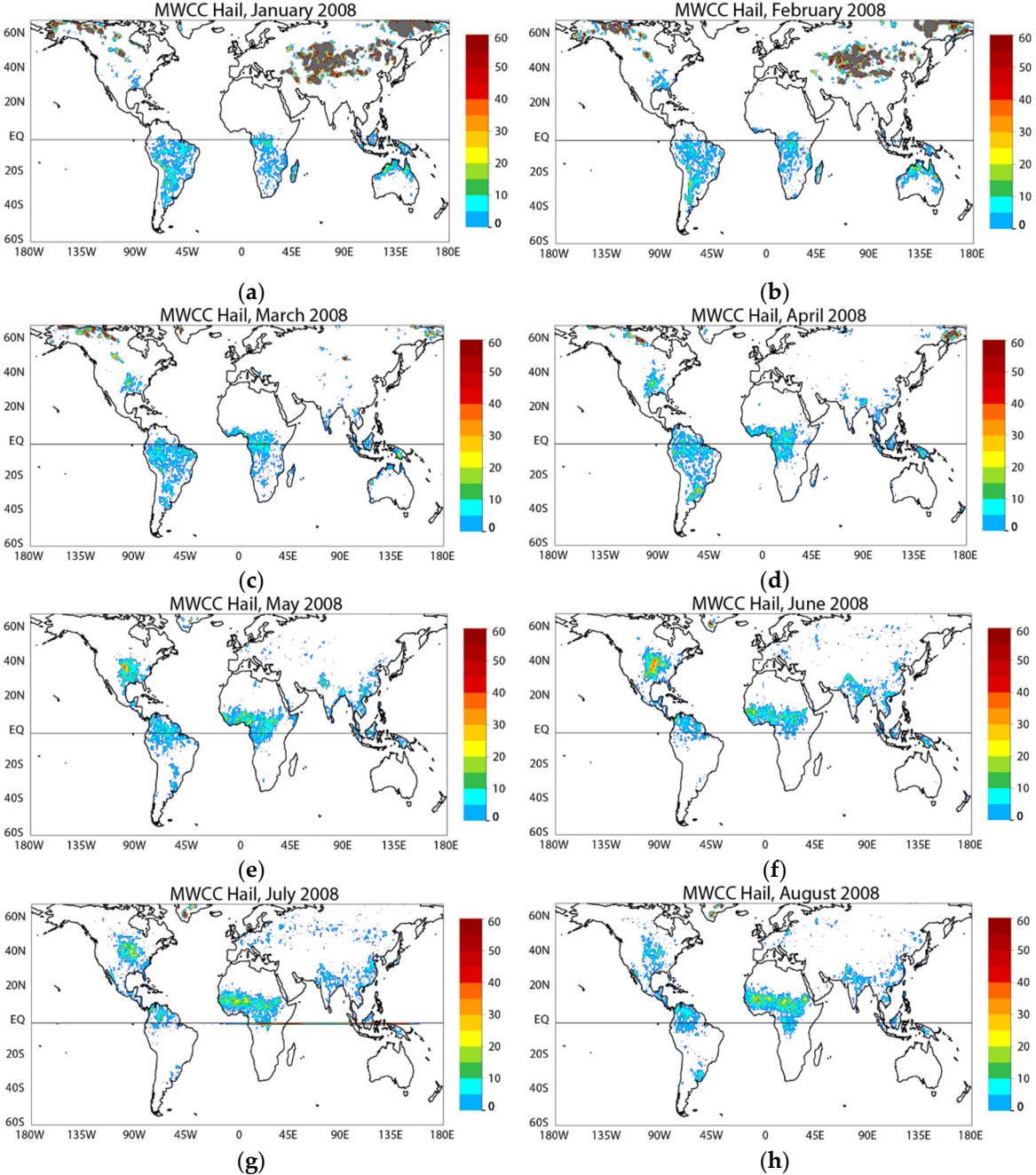

**Figure 10.** *Cont.*

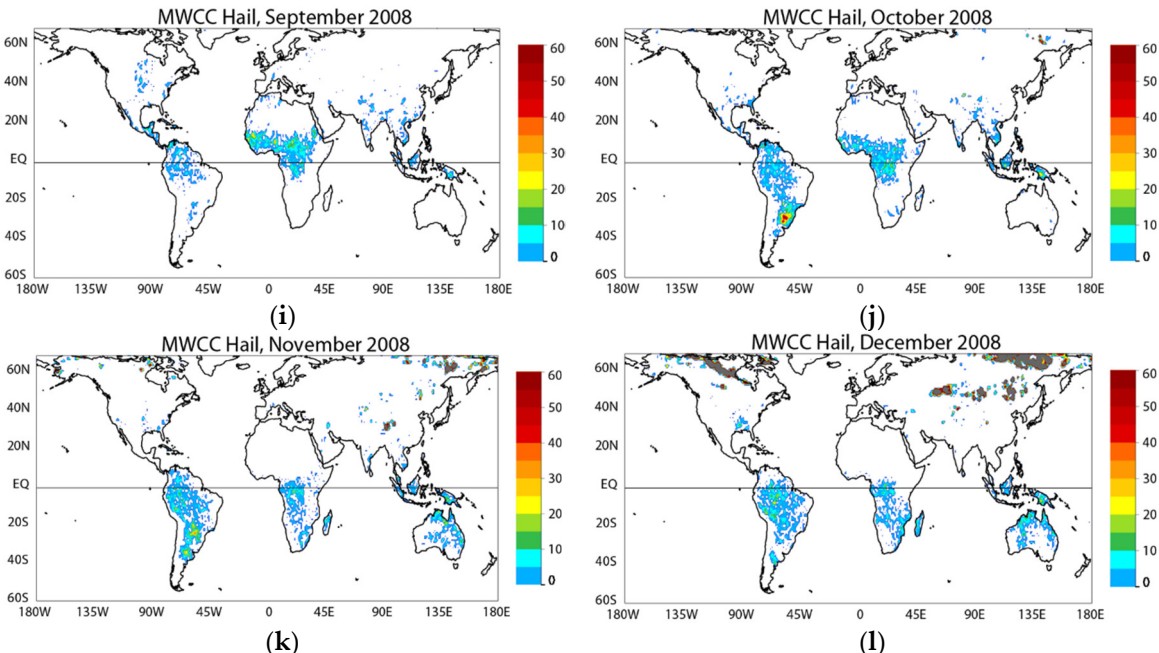

**Figure 10.** Monthly global distribution of hail events for the year 2008 January–December (**a**) to (**l**) as detected by the MWCC-H. The total monthly hail occurrences are aggregated within a 1° grid box. Brown areas are due to cold and/or high elevation terrains, where the algorithm cannot distinguish between surface and hail.

## 5. Discussion and Conclusions

The aim of this work is to propose a new method for observing hail clouds from space using the frequency range 150–170 GHz. The main differences from previous methodologies for detecting hail are (a) using only high frequencies the model captures the signal from small hail/graupels to very large hail; (b) using for the first time a compact and flexible model to detect hail we get homogeneous hail maps from the whole GPM-C; (c) exploiting the possibility of using this fast running method for hail detection makes it possible operational implementations in nowcasting processing chains; and (d) this approach can be effective when applied to the future satellite microwave radiometers covering the proposed frequency range.

A prior sensitivity analysis demonstrated the potential of frequencies in the range 150–157 GHz to effectively identify hail signatures by distinguishing the PMW signal depression from a few millimeter to very large hail diameters. The high dynamic sampling range offered by these frequencies associated with the need for developing a flexible and fast model for defining hail patterns has driven the authors to use a modified logistic growth model for calculating the hail probability coupled to the hail diameter. The MWCC-H proposed by Laviola et al. [20] satisfied the expectations demonstrating high performances in detecting hail and a high degree of adaptation to the frequency range 150–170 GHz that characterize the GMI.

The first experiments showed that the MWCC-H, originally developed for AMSU-B/MHS, was able to detect hail clouds when directly applied to the GPM-CO. However, in order to reduce possible radiometric discrepancies between the reference instrument (MHS) and the MHS-like sensors, an adjustment of the GPM-CO sampling channels in the range 150–170 GHz was required. The calculation of the fitting coefficients helped to rearrange the probability values from the GPM-CO in the score range of the MHS by avoiding crossing the probability value 1 as experienced when the MWCC-H was directly applied to the GPM-CO.

The rationale of this approach is then to extend the MWCC-H model to all MHS-like radiometers orbiting on the GPM-C. The adjustment of the MHS-like frequency channels in the range 150–170 GHz on the MHS frequency channel at 157 GHz allowed to compute the hail probability using the

original MWCC-H equation but fed with the MHS-like adjusted TBs. A suite of correction terms has been calculated in order to minimize errors reaching the convergence between the original and the generalized MWCC-H model. Thus, a collection of the T-H proxy curves between TB and hail probability was calculated. The T-H diagrams, where H increases and the temperature T decreases correspondingly, represent a proxy for quickly identifying the hail category through the TBs observed from each sensor. The steepness of the T-H relationship works as a sort of "slide rule" to classify the severity of the hailstorm by progressively detecting the increase of the hail probability associated with the hail diameter.

The results show the general good performance of MWCC-H when applied to the complete suite of GPM-C sensors. As demonstrated in Section 3, the quasi near-real time reconstruction of the sequence of hailstorms on the Adriatic Sea speaks in favor of the strength of this approach to realistically reproduce the evolution of hailstorms. Thus, the application of the MWCC-H on a long-period dataset can be a robust tool to draw the study the seasonality of hail events at the global scale as shown in Section 4, where the global annual map of hail events from AMSU-B is presented. Further applications were also useful for reconstructing the regional climatology of hail events in the Mediterranean basin as seen from the GPM-C. This analysis provided the distribution of hail events over 20 years by highlighting the most affected areas of the Mediterranean basin. The analysis is on its way and results will be presented soon.

The first applications were recently experimented on test data of the new microwave sounder (MWS) sensor to be launched onboard the EUMETSAT Polar System-Second Generation (EPS-SG). The MWS has a direct heritage from the microwave instruments AMSU-A and MHS with the addition of two temperature and three humidity sounding channels and a new channel at 229 GHz to provide information on cirrus clouds by improving humidity sounding information. The synthetic dataset used in our investigation was released by EUMETSAT for the user community in their preparations for receiving and processing the EPS-SG data. Our tests allowed us to calculate a primary dataset of level 2 products in terms of instantaneous precipitation through the 183-WSL algorithm [29,30] and cloud and hail products using the MWCC-H. The results were also useful to encourage further ongoing experiments with other new sensors equipping the EPS-SG mission that will be operational from 2021.

Finally, the results of our applications also stimulated thoughts on new challenging satellite missions such as those at the geostationary orbit satellite equipped with a high-frequency multi/hyperspectral microwave sensor (HYMS). The results of Section 3, where a sequence of hailstorms has been detected at very high time frequency, suggest that a geostationary PMW sensor would significantly change the way of observing meteorological phenomena by providing full hemispheric observations with an accuracy very close to that of polar-orbiting radiometers. Although the discussion has been on the table for almost 20 years, and in spite of the substantial recent technology jump forward, the microwave sounding from GEO still represents the new frontier of the Earth Observation from space. The interest of the scientific community in exploring Earth with the obvious benefits of continuously monitor the same portion of the Earth during all weather conditions encouraged specific studies for equipping GEO satellites with a HYMS. The hyperspectral observation at MW frequencies makes it possible to determine the vertical distribution of temperature and humidity of the atmosphere with a large number of densely spaced weighting function in the vertical. Thus, the very high temporal resolutions of the GEO systems combined with the high spectral resolution of HYMS points to improve nowcasting and very short-range forecasts by making available satellite methodologies currently used for inferring the cloud structure and hydrometeor properties only from polar orbiters.

**Author Contributions:** S.L. conceived the method and carried out the data analysis together with G.M., V.L. contributed to the concept. R.R.F. and J.B. contributed to the validation. All authors contributed to the writing. All authors have read and agreed to the published version of the manuscript.

**Funding:** This research received no external funding.

**Acknowledgments:** This work is performed in the context of the scientific cooperation defined by the MOU between the Institute of Atmospheric Sciences and Climate of the Italian National Research Council (CNR-ISAC) and the National Oceanic and Atmospheric Administration (NOAA).

**Conflicts of Interest:** The authors declare no conflict of interest.

## Abbreviations

| | |
|---|---|
| AMSR-E | Advanced Microwave Scanning Radiometer for the Earth Observing System |
| AMSU-A | Advanced Microwave Sounding Unit-A |
| AMSU-B | Advanced Microwave Sounding Unit-B |
| ATMS | Advanced Technology Microwave Sounder |
| CNES | Centre National d'Etudes Spatiales |
| CNR | Consiglio Nazionale delle Ricerche |
| DMSP | Defense Meteorological Satellite Program |
| DPR | Dual-frequency Precipitation Radar |
| EFOV | Effective FOV |
| EPS-SG | EUMETSAT Polar System-Second Generation |
| ESWD | European Severe Weather Database |
| EU | Europe |
| EUMETSAT | European Organization for the Exploitation of Meteorological Satellites |
| FOV | Field of View |
| GEO | GEostationary Orbit |
| GMI | GPM Microwave Imager |
| GPM | Global Precipitation Measurement mission |
| GPM-C | GPM Constellation |
| GPM-CO | GPM Core Observatory |
| HYMS | HYperspectral Microwave Sensor |
| IFOV | Instantaneous FOV |
| IR | InfraRed |
| ISAC | Istituto di Scienze dell'Atmosfera e del Clima |
| ISRO | Indian Space Research Organisation |
| JAXA | Japan Aerospace Exploration Agency |
| MHS | Microwave Humidity Sounder |
| MOA | MetOp A |
| MOB | MetOp B |
| MWCC | MicroWave Cloud Classification method |
| MWCC-H | MicroWave Cloud Classification method for Hail detection |
| MWS | MicroWave Sounder |
| NEXRAD | Next Generation Radar network |
| NOAA | National Oceanic and Atmospheric Administration |
| NPP | National Polar-orbiting Partnership |
| PMW | Passive Microwave |
| PMWCC | Perturbation of MWCC |
| SPC | Storm Prediction Center |
| SSM/I | Special Sensor Microwave/Imager |
| SSMIS | Special Sensor Microwave Imager/Sounder |
| TB | Brightness Temperature |
| T-H | Temperature-Hail proxy curve |
| TMI | TRMM Microwave Imager |
| TRMM | Tropical Rainfall Measuring Mission |
| US | United States of America |
| UTC | Universal Time Coordinated |
| VIS | Visible |
| 183-WSL | Water vapor Strong Lines at 183 GHz algorithm |

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
