# Peer review of "A New Method for Hail Detection from the GPM Constellation: A Prospect for a Global Hailstorm Climatology"

_remotesensing, doi:10.3390/rs12213553_

Round 1
Reviewer 1 Report
Review of "A method for hail detection from the GPM constellation and a prospective for a global hailstorm climatology" by Laviola et al.
This paper presents a method for detecting hailstorms using satellite radiometers in the 150-170 GHz frequency range. The algorithm adjusts these values to a reference 157 GHz instrument and uses these adjusted values to compute a probability of hail as the final product. The probability of hail product is then shown to correctly identify hail in handful of chosen storms and, additionally, a global climatology is calculated for each month of the year (albeit from just 1 year of data: 2008).
I am not convinced that this paper contributes significant value to be worthy of publication. I have several significant concerns that would need addressing before considering publication.
1) The motivation for doing this work is not discussed. There are other attempts to produce hail climatologies, some from satellite brightness temperatures (as in this paper), others from overshooting tops and others from radar or surface observations. Some of these studies are mentioned in the introduction, but and weaknesses in them and therefore the motivation for the current work is completely missing. As a reader I am left to think "why have you done this study?"
2) The method does not appear to be new, but rather a refinement/adaptation to new instruments of the method presented in Laviola et al. 2020 (ref 20)
Despite claims that "A new method for detecting hailstorms by using all the MHS-like satellite radiometers currently in orbit is presented." (abstract; line 12), this is later clarified. "The study presented here aims to extend the potential of the MWCC-H hail detection model proposed by Laviola et al. [20] to the entire GPM-C" (introduction, line 73). Furthermore, "The model for hail detection presented in Laviola et al. [20] ... has been directly employed" (introduction, line 91).
As a reader I am left to think "is it really new?"
3) Existing work, e.g. Cecil and Blankenship [14], uses frequencies of 36.5 and 89 GHz to perform a very similar analysis. The motivation for choosing different frequencies is never explained.
As a reader I am left to think "what is different from previous studies?"
4) The analysis of the case studies and especially climatologies is of bare minimum quality - the climatology isn't even compared with other existing climatologies.
As a reader I am left to think "how well do these results compare to other climatologies?"
5) The climatology presented is only for a single year (this could be acceptable if it weren't for the other defecincies in the paper).
As a reader I am left to think "are the results reproducible?"
6) The physics behind why the method works and its limitations are never discussed. As a reader I am left to think "how does this method really work?"
7) Too many acronyms to understand as a non-specialist (one of many examples, lines 613-616)
Another example - where the MWCC-H acronym is first introduced, no definition of the acronym is giving, making it rather hard to remember throughout the rest of the article.
8) The "high performances of the hail model" (abstract, line 20) stated in the abstract is never really examined. Interrogation of the computed climatologies and comparison to other climatologies is completely missing. As a reader I am left to think "does this method really produce good results?"
9) The performance of this algorithm seems to be poor in some locations, for example the tropics("Further considerations can be drawn on the tropical regions where the number of detect hail events is probably unrealistic." Why is the method unreliable in the tropics?"; Line 711-713 ). The judgement of these values being probably unrealistic, and the cause of the failings are never discussed. As a reader I am left to think "what is the cause of these bad results?"
As a summary of my feelings having read to the end of the paper:
"why have you done this study?"
"what is different from previous studies?"
"is it really new?"
"how does this method really work?"
"how well do these results compare to other climatologies?"
"does this method really produce good results?"
"what is the cause of the bad results?"
"are the results reproducible?"
Due to all of these open questions, I cannot recommend that the article be published without significant changes.
Author Response
Please, see the attached file.

Reviewer 2 Report
Review of 'A method for hail detection from the GPM constellation and a prospective for a global hailstorm climatology', Laviola et al.
In this study, authors developed a new method for detecting hailstorms by using all the MHS-like satellite radiometers. This method takes into account the differences in the sampling frequency, the scanning mechanism and the spatial resolution between different sensors. This method is applied globally in the last section to analyze the geographic and seasonable variabilities. I would recommend to accept with minor revisions.
- In table 1, different polarized channels have been used in the algorithm (e.g., V for GPM, H for SSMIS). Is this proposed method sensitive to the different polarized channels?
- Line 155: The selected cases from each satellite pair have to have overpasses within 30 min, but sometimes isolated convection can have a very rapid lifecycle. How much is the result (hail probability) sensitive to this threshold?
- Figure 9: I like the idea of showing the results from different sensors, but the colors on this plot is very different to see. I would recommend to zoom in on the map and adjust the colorbar.
Author Response
Please, see the attached file.

Reviewer 3 Report
REVIEW of the paper "A method for hail detection from the GPM constellation and a prospective for a global hailstorm climatology" by Sante Laviola, Giulio Monte, Vincenzo Levizzani, Ralph R. Ferraro and James Beauchamp [Remote Sens. 898879-peer review].
This manuscript deals with extremely current topic since hail is an extreme meteorological phenomenon in mid-latitudes. A new method for detecting hailstorms by using all the MHS-like satellite radiometers currently in orbit is presented. A probability-based model originally designed for AMSU-B/MHS has been fitted to the observations of all microwave radiometers onboard the satellites of the GPM constellation. All MHS-like frequency channels in the 150-170 GHz frequency range were adjusted on the MHS channel 2 (157 GHz). The hail detection model to the whole GPM constellation demonstrates the high potential of this generalized model to map the evolution of hail-bearing systems at very high temporal rate. The results on the global scale are successful and promising in a sense of further improvement.
Comments
The manuscript is well written and well organized in 5 sections. It is richly illustrated by 10 Figures and 11 Tables. In my opinion, this paper presents an original contribution to hail detection methods and hail climatology. I would therefore welcome its publishing in this eminent journal. The use of satellites seems to be the promising approach in hail detection in future. However, I would like to ask a few questions regarding the topic of the manuscript as follows:
- What would be the estimated error of this method?
- Hailstorm splitting is one of the very important process. What are the possibilities of this method in the detection of this process?
- Can this method be used fornowcasting?
Author Response
Please, see the attached file.
